# Investigating the Potential of Cosmic-Ray Neutron Sensing for Estimating Soil Water Content in Farmland and Mountainous Areas

Yifei Jiang [1,2], Kefan Xuan [1,2], Chen Gao [1,2], Yiren Liu [1,2], Yuan Zhao [3], Haodong Deng [4], Xiaopeng Li [1,*] and Jianli Liu [1,5,*]

1   Institute of Soil Science, Chinese Academy of Sciences, Nanjing 210008, China; jyf@issas.ac.cn (Y.J.); kfxuan@issas.ac.cn (K.X.); gaochen@issas.ac.cn (C.G.); liuyiren@issas.ac.cn (Y.L.)
2   University of Chinese Academy of Sciences, Beijing 100049, China
3   College of Environmental & Resource Sciences, Shanxi University, Taiyuan 030006, China; zhaoyuan383@163.com
4   College of Hydrology and Water Resources, Hohai University, Nanjing 211100, China; ddenghaodong1997@163.com
5   University of Chinese Academy of Sciences, Nanjing 211135, China
*   Correspondence: lixp@issas.ac.cn (X.L.); jlliu@issas.ac.cn (J.L.)

**Abstract:** The conventional methods of estimating soil water content (SWC) are mainly based on in situ measurements at sampling points and remote sensing measurements over an entire region. In view of these methods, cosmic-ray neutron sensing (CRNS) has received increasing attention in recent years as a mesoscale, noncontact SWC estimation technology that can provide more accurate and timely estimates of SWC over a larger area. In this study, we estimated SWC using both CRNS and soil-mounted detectors in farmland and mountainous areas, and evaluated the accuracy of the estimations at two experimental sites. Ultra-rapid adaptable neutron-only simulation (URANOS) was used to simulate the detection radius and depth of the two experimental sites and to obtain the spatial weights of the CRNS footprint. The results show that the theoretical range of detection was reduced in farmland compared to mountainous areas during the experimental period, suggesting that farmland retained more SWC even with less precipitation. Spatial weights were simulated to calculate the SWC of sampling points, and the weighted and averaged SWC were then correlated with CRNS. The weighting calculation improves the accuracy of CRNS estimations, with a determination coefficient ($R^2$) of 0.645 and a root mean square error (RMSE) of 0.046 $cm^3 \cdot cm^{-3}$ for farmland, and reproduces the daily dynamics of SWC. The $R^2$ and RMSE in mountainous areas are 0.773 and 0.049 $cm^3 \cdot cm^{-3}$, respectively, and the estimation accuracy of CRNS cannot be improved by the weighting calculation. The estimation accuracy of CRNS is acceptable in both regions, but the mountainous terrain obstructs neutron transmission, causing a deviation between the actual and theoretical neutron footprints in mountainous areas. Thus, the accuracy of SWC estimation is limited in mountainous terrain. In conclusion, this study demonstrates that CRNS is suitable for use in farmland and mountainous areas and that further attention should be given to the effects of topography and vegetation when it is applied in mountainous environments.

**Keywords:** cosmic-ray neutron sensing; soil water content estimation; farmland; mountainous terrain

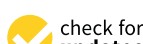



## 1. Introduction

Soil water content (SWC) is an important factor in research in agriculture, forestry, hydrology, and other fields. Many methods are available for determining SWC, including time domain reflectometry (TDR) and frequency-domain reflectometry (FDR) [1,2] for point-scale SWC determination, and satellite-based remote sensing [3,4] for regional or global SWC estimation. The measurement scales of these methods are either a few centimetres or

a few kilometres. Cosmic-ray neutron sensing (CRNS) has shown promise for filling the gap between the point scale and remote sensing scale in recent studies [5,6].

The use of cosmic-ray neutron sensing to determine SWC was first proposed by Zreda et al. [7] during preliminary research on cosmic-ray neutron transport [8,9]. The theory of CRNS can be described as follows: fast neutrons with medium energy near the land surface are created by cosmic rays interacting with the atmosphere. Fast neutrons experience significant energy loss and are converted into thermal neutrons when reacting with hydrogen nuclei. Therefore, the intensity of fast neutrons is inversely correlated to the hydrogen atom intensity near the ground surface. The fast neutron intensity detected by CRNP (cosmic ray neutron probe) can indicate the hydrogen content near the ground surface. The hydrogen content near the ground surface is the main component of soil water content, which can be calculated from the measured fast neutron density.

As a passive detection method, the detection area of CRNS is determined by the neutron footprint, which is defined as the area around the probe from which 86% of counted neutrons arise. The maximum footprint of CRNS at sea level is 335 m in radius according to the calculation of Zreda et al. [7] using Monte Carlo N-Particle transport code (MCNP), and the measurement depth ranges from 0.76 m in dry soils to 0.12 m in wet soils. MCNP simulations explain the theoretical footprint and measurement depth of CRNS, but the neutron footprint in the horizontal direction is correlated with neutron intensity, which is affected by atmospheric pressure, altitude, and other hydrogen sources [10]. A deviation occurs between the theoretical neutron footprint and the actual measured footprint when using CRNS under different ecological conditions [11]. Köhli et al. [12] considered the influencing factors and improved the footprint simulations via the Monte Carlo code ultra rapid adaptable neutron-only simulation (URANOS). These simulations indicate that the revised footprint radius is 240 m at sea level, and the measurement depth ranges from 0.15 m to 0.83 m under different soil moisture conditions. The simulation results of URANOS also indicate that the signal strength per radial distance is highly nonlinear and can be described as a weighting function.

The revised footprint simulated with URANOS has proven compatible with field studies of CRNS and has been applied in several studies [5,13–15]. However, the application of CRNS in different scenarios still requires specific calculations. The most suitable research sites for CRNS are flat terrains with relatively dry climates, which provide stable signals for CRNP. Therefore, CRNS is more widely used in farmland and agro-pastoral areas [16–18]. Forest ecosystems provide a wide variety of hydrogen pools, creating a worst-case scenario [19]. Heidbüchel et al. [20] suggested that several factors, such as the presence of a litter layer and spatially-heterogeneous soil moisture conditions within the sensor footprint, could cause inaccurate estimations of the footprint when applying CRNS to forests. Currently, research on the applicability of CRNS in humid forest conditions is limited.

In flat terrain conditions, CRNS has been used to measure field-scale root zone soil moisture [21], aboveground water equivalent [22], the retrieval of soil ice content [23], etc. However, open questions remain regarding how to account for spatial heterogeneity. Previous studies have suggested that neutrons can travel tens of centimetres in soil with the respective moisture conditions, beyond which distance they will be absorbed [24]. This finding implies that a portion of the neutrons are absorbed by the soil when they traverse intricate terrain. The simulation of the CRNS detection range is based on the assumption that the underlying surface is flat in the horizontal direction. The discrepancy between the simulated and actual detection ranges of CRNS may lead to uncertainty in the estimation. In contrast, the accuracy of CRNS estimation in complex mountainous terrain needs further study.

Therefore, we have conducted the present experiment to achieve three objectives: (i) investigate the potential of CRNS for estimating SWC in farmland and mountainous areas, (ii) examine the characteristics of CRNS footprints in farmland and mountainous

areas, and (iii) explore the influence of terrain and vegetation factors on the accuracy of CRNS estimation.

## 2. Experimental Site and Instrumentation

To study the applicability of CRNS under different ecological conditions, two experimental sites were established to represent the typical environmental conditions of the Huang-Huai-Hai Plain in northern China and the Taihu Basin in southeastern China. The aim of this research is to evaluate the effectiveness of CRNS in estimating SWC in flat farmland areas with a warm-temperate, semihumid monsoon climate, and steep headwater hillslopes in a subtropical humid climate.

### 2.1. Cosmic-Ray Neutron Sensing Probe and Meteorological Equipment

We developed a cosmic-ray neutron sensing probe (CRNP) for this study using a proportional counter tube filled with $BF_3$, which can detect neutrons in the fast energy range. The fast neutron is absorbed while passing through the tube and induces a pulse of electrical current that is sent to the pulse module. Then a counting module (CR300, Campbell Scientific, Logan, UT, USA) records an electrical pulse signal that is proportional to the neutron density (Figures 1 and 2).

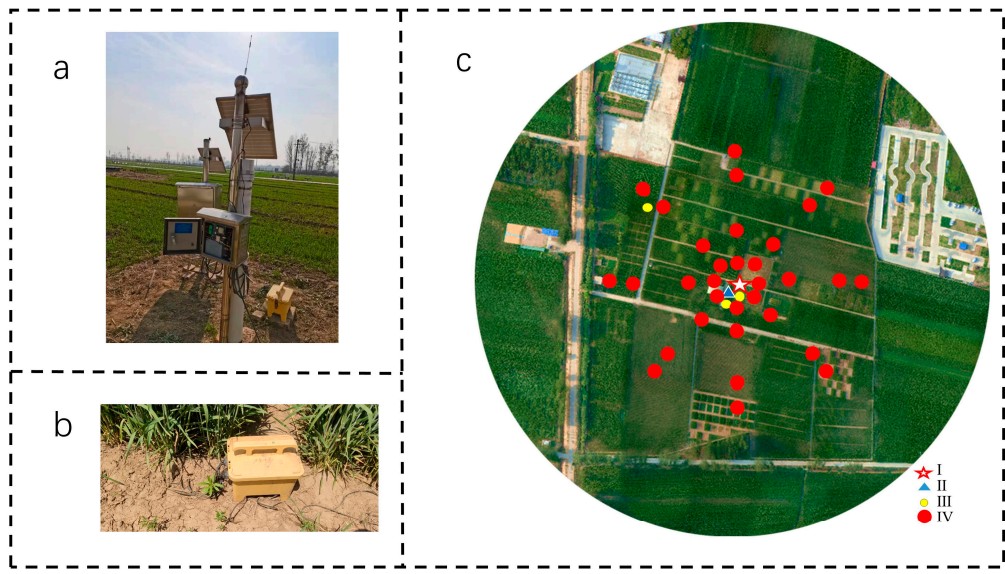

**Figure 1.** Location of the cosmic-ray neutron sensing probe and the experimental sampling design at Fengqiu station. (**a**) Cosmic-ray neutron sensing probe. (**b**) Frequency domain reflection (FDR) auto soil moisture monitor. (**c**) Bird's-eye view of the study site, I: cosmic-ray neutron sensing probe; II: meteorological instrument; III: FDR auto soil moisture monitors; IV: soil sampling points.

Meteorological observation equipment (H21-USB, Bourne, MA, USA) was installed near the CRNP to record atmospheric pressure, atmospheric temperature, and air humidity data. At the same time, local precipitation data were recorded by the equipment, and all meteorological data were recoded once an hour (Figures 1 and 2).

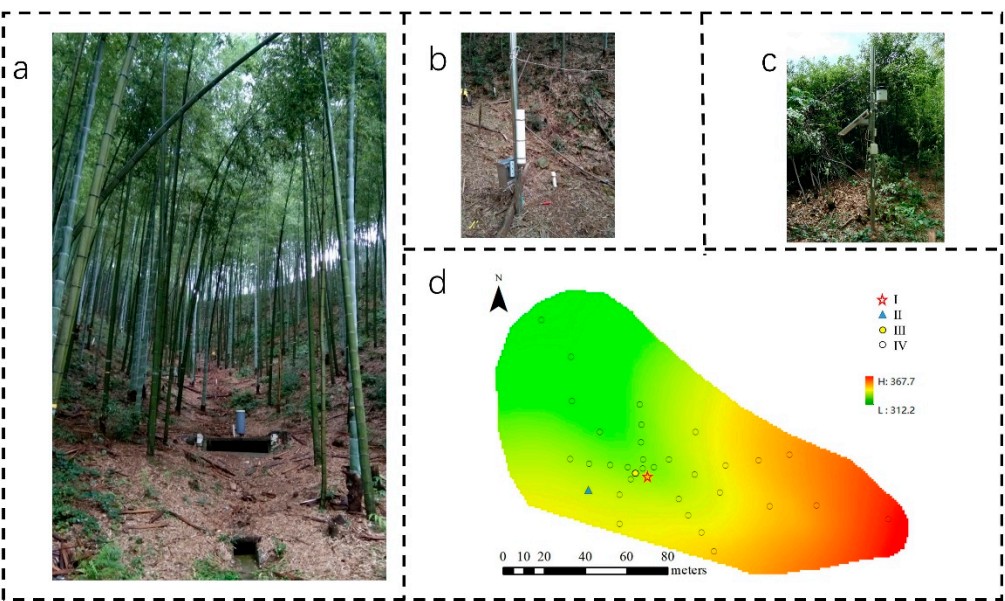

**Figure 2.** Location of the cosmic-ray neutron sensing probe and the experimental sampling design at Hemuqiao station. (**a**) View of the study site. (**b**) CRNP system. (**c**) Meteorological instrument. (**d**) Digital elevation map and sampling points of the study site, I: CRNP; II: Meteorological instrument; III: FDR auto soil moisture monitor; IV: Soil sampling points.

### 2.2. Study Site 1

The study was conducted at the Fengqiu State Key Agro-Ecological Experimental Station, Fengqiu Country, Henan Province, China (114°34′ E, 35°01′ N). The elevation at the Fengqiu station is 67 m above the mean sea level. The site has a typical monsoon climate at medium latitudes with annual temperatures averaging 13.9 °C and annual precipitation averaging 605 mm, 50~90% of which occurs from May to October. The field has been cultivated with winter wheat and summer maize, two crops in a 1-year rotation, for at least 50 years.

For this study site, the CRNP was installed in the middle of the Fengqiu Station (Figure 1). Soil samples were taken from 32 points at radial distances of 1, 25, 50, 100, and 125 m from the CRNP in eight directions. Samples were taken using a soil corer 80 cm in length and 5 cm in diameter, which enables samples to be divided into 8 layers with 10 cm-long soil columns.

To determine the continuous change in soil moisture, three FDR auto soil moisture monitors with a determination frequency of once per hour were installed at distances of 1, 25, and 100 m from the CRNS probe. The FDR monitors were calibrated by the oven-drying method.

### 2.3. Study Site 2

A headwater hillslope with an area of 0.42 ha was selected for the study, which was conducted at the Hemuqiao experimental station in southeast China (119°47′ E, 30°34′ N). The elevation in Hemuqiao ranges from 312 to 367 m above mean sea level. The study site has hydrologic characteristics typical of the humid hilly areas of southeast China, with an average annual precipitation of approximately 1580 mm. The site's average annual temperature and evapotranspiration are 14.6 °C and 805 mm, respectively. The vegetation type is moso bamboo (Phyllostachys edulis), which covers all areas of the hillslopes. The shape of the hillslope is similar to a saddle, with a valley in the middle and two hillsides on both sides. The slope of the hill can reach 30.5°, and the soil depth ranges from 13 cm to 171 cm, with uneven distribution.

At this study site, the CRNP was installed in the middle of the hillslope, and soil samples were taken from 31 points in eight directions of the CRNP (Figure 2). Samples were

taken using a soil corer 30 cm in length and 5 cm in diameter, which enables the sample be divided into 3 layers with 10 cm-long soil columns. At the first sampling, soil moisture was measured by a TDR monitor, and soil samples were simultaneously collected. Then the soil columns were oven-dried to determine their SWC and calibrate the TDR. Due to topographical factors, the TDR soil moisture monitor was used in subsequent soil moisture sampling of this study site. Meanwhile, an FDR auto soil moisture monitor was installed near the CRNP to determine the rate of continuous change in soil moisture at study site 2.

## 3. Methodology

### 3.1. Calibration

Cosmic ray neutron sensing probes (CRNPs) are particle detectors that measure the neutron intensity near the land surface [25]. The CRNP neutron counts were affected by atmospheric pressure, air humidity, and incoming neutron flux.

Atmospheric pressure determines the probability that fast neutrons will collide with particles while moving through the atmosphere and produce energy decay. Therefore, atmospheric pressure affects the density of fast neutrons near the land surface. The pressure correction factor $f_p$ is given by [25]:

$$f_p = \exp\frac{P - P_0}{L} \tag{1}$$

where $L$ is the mass attenuation length for high-energy neutrons (g·cm$^{-2}$), which was determined to be 138 g·cm$^{-2}$ in this study; $P$ is the pressure at the specific site; and $P_0$ is an arbitrary reference pressure (the long-term average pressure at the specific site, KPa).

Air humidity indicates the atmospheric water vapour content, which affects the density of fast neutrons near the land surface. The atmospheric water vapour correction factor $f_w$ is defined as [26]:

$$f_w = 1 + 0.0054(\rho - \rho_0) \tag{2}$$

where $\rho$ is the absolute humidity at the time of measurement (g·m$^{-3}$) and $\rho_0$ is the absolute humidity at the reference time (g·m$^{-3}$).

Fast neutron intensity near the land surface is sensitive to temporal changes in incoming neutron flux, which should be corrected with the factor $f_i$ and can be expressed as [25]:

$$f_i = \frac{I_m}{I_0} \tag{3}$$

where $I_m$ is the measured neutron intensity of the cosmic-ray neutron monitors, which are designed to detect high-energy secondary neutrons. $I_0$ is a specified baseline reference intensity at a given time. The neutron monitor at Yang-Ba-Jing, China was used in our study.

The CRNP neutron counts were then corrected with the three factors given above in the form of:

$$N = N_m \times f_p \times f_w / f_i \tag{4}$$

where $N$ is the calibrated CRNP neutron count and $N_m$ is the measured neutron count from CRNP.

### 3.2. Estimation of SWC

The function for converting neutron count into SWC was improved by Desilets et al. [11] and is expressed as:

$$\theta = \frac{\rho_{bd}}{\rho_w}\left(\frac{a_0}{N/N_0 - a_1} - a_2\right) \tag{5}$$

where $\theta$ is the volumetric water content (cm$^3$·cm$^{-3}$), $\rho_{bd}$ is the soil bulk density, $\rho_w$ is the water density, $N$ is the neutron count calibrated with Equation (4), $N_0$ is the counting rate over dry soil under the same reference conditions, and $a_i$ is the fitting parameter. The

parameters were defined as $a_0 = 0.0808$, $a_1 = 0.372$, and $a_2 = 0.115$ for water contents higher than 0.02 kg kg$^{-1}$.

### 3.3. Horizontal Weight

Currently, the most reliable function for describing the spatial weighting of CRNS is the weighting function obtained from the neutron transport distribution fitted by Monte Carlo simulations. The Monte Carlo-based URANOS also provides a simpler method to obtain neutron transport distributions without fitting the weighting function.

In this study, URANOS was used to obtain the neutron transport distribution curves at the two sites. Then, the distribution curves were integrated to determine the contribution weight of the CRNS soil sampling points. This method simplifies the step of fitting the neutron transport distribution curve into the weighting function and avoids overfitting errors. URANOS accounts for factors affecting neutron transport, such as air humidity and soil bulk density, resulting in simulation weights that more closely approximate true weights.

### 3.4. Vertical Weight

The penetration depths of the detected neutrons for different distances from the CRNP proved to be different. Therefore, the conventional vertical weighting function, which used a linear relation from Franz et al. [24], needed to be improved. In this work, we used the revised vertical weighting function given by Schrön et al. [13] and expressed it as:

$$W = exp\frac{-2d}{D} \tag{6}$$

where $W$ is the weight at different depths, $d$ (cm). $D$ denotes the effective penetration depth, defined as the depth within which 86% of neutrons probed the soil. The value of $D$ for different distances can be calculated by Schrön et al. [13].

### 3.5. Estimation of Measurement Accuracy

The measurement accuracy of CRNS was quantified using the linear correlation between the SWC and CRNS determination results. Calculation indexes include the coefficient of determination ($R^2$) and root mean square error (RMSE). The $R^2$ and RMSE are given by:

$$R^2 = \frac{\sum_{i=1}^{n}\left(x_i - \bar{x}\right)\left(y_i - \bar{y}\right)^2}{\sum_{i=1}^{n}\left(x_i - \bar{x}\right)^2 \sum_{i=1}^{n}\left(y_i - \bar{y}\right)^2} \tag{7}$$

$$\text{RMSE} = \sqrt{\frac{1}{n}\sum_{i=1}^{n}(x - y)^2} \tag{8}$$

where $n$ is the number of data points and $x$ and $y$ are the SWCs measured by the point samples and CRNS, respectively.

## 4. Results and Discussion

### 4.1. Footprint Estimation at Different Study Sites

Footprint and weighting functions show good applicability in the farmlands of Fengqiu due to their flat topography. However, the CRNP at the Hemuqiao station had an unascertained footprint because of its complex terrain. Therefore, we conducted multiple simulations under different air humidity and soil moisture conditions to determine the variation in the neutron footprint under different ecological conditions.

The intensity of cosmic-ray neutrons depends on the hydrogen content above the ground. In our simulations, the maximum detection radius and depth were obtained in the driest condition of the study sites. In contrast, the minimum detection radius and depth were obtained in the wettest condition of the study sites. The maximum and minimum

detection depths in Fengqiu were 31.2 cm and 16.9 cm, respectively, both of which were shallower than those of Hemuqiao (see Table 1). In Franz et al.'s research [24], the detection depth was primarily determined by soil hydrogen sources, which included surface water, SWC, and lattice water, indicating that the soil hydrogen source reserve is higher in Fengqiu than in Hemuqiao.

**Table 1.** The footprint of the CRNS simulated by URANOS at different study sites. The theoretical values of the detection radius and detection depth were obtained from simulation results under extremely dry and extremely humid conditions.

| Study Site | Maxim Radius | Minimum Radius | Maxim Depth | Minimum Depth |
|---|---|---|---|---|
| Fengqiu | 139 m | 127 m | 31.2 cm | 16.9 cm |
| Hemuqiao | 153 m | 125 m | 45.9 cm | 19.1 cm |
| Theoretically | 218 m | 123 m | 65.9 cm | 13.8 cm |

Our simulation indicated that the detection radius was more significantly affected by air humidity than by SWC (see Appendix A). However, no obvious difference in detection depth was found under different air humidity conditions (see Appendix B). We assumed that the detection depth at a specific point was affected only by its SWC. The two study sites had different maximum and minimum detection depths due to the difference in soil water conditions. Fengqiu is an irrigated agricultural field that provides a vast quantity of hydrogen, including crops and soil water. In contrast, the terrain of Hemuqiao is a mountain slope, where the soil cannot retain water for a long time due to the steep gradient and uneven distribution of soil depth (see Section 2.3). Therefore, the maximum and minimum detection depths of Hemuqiao were larger than those of Fengqiu.

According to previous studies, the effective detection radius [7,24,26,27] is influenced by a combination of factors, including SWC, atmospheric pressure, water vapour, and aboveground biomass. As acknowledged by previous studies, the horizontal footprint is determined mainly by atmospheric density and humidity [25]. Hemuqiao has a higher altitude than Fengqiu, and its soil hydrogen inventory is less than that at Fengqiu during the driest period, resulting in a maximum detection radius of 153 m, which was larger than the detection region of Fengqiu. However, the minimum detection radii of the two study sites were comparable due to their complex meteorological and vegetation conditions. First, Hemuqiao had a higher average annual precipitation than Fengqiu. Second, a litter layer and moso bamboo had an interception effect on precipitation, which resulted in higher air humidity at Hemuqiao during the precipitation period. The combination of these factors resulted in similar minimum detection radii at Hemuqiao and Fengqiu, which were 125 m and 127 m, respectively.

### 4.2. SWC Variation and Seasonal Characteristics at Different STUDY Sites

The time series of SWC at the two sites are shown in Figure 3. The soil water measured by FDR was calibrated with the oven drying method; hence, it could represent the actual SWC time series (blue dotted line). In this research, SWC measured by the CRNS had a similar trend to the SWC time series during the growing season. This variation pattern was observed at both study sites, indicating that CRNS can characterize the time series of SWC. SWC measured by the CRNS was inversely proportional to the neutron intensity. The detected neutrons interacted with the hydrogen in the footprint and carried information about its hydrogen inventory [7]. The SWC ranged from 0.10 $cm^3 \cdot cm^{-3}$ to 0.40 $cm^3 \cdot cm^{-3}$ at the Fengqiu station and from 0.05 $cm^3 \cdot cm^{-3}$ to 0.30 $cm^3 \cdot cm^{-3}$ at the Hemuqiao station (see Figure 3). As expected, the range of SWC and the footprint simulations in Section 4.1 consistently showed that Fengqiu stored a larger amount of SWC during the experimental periods. In general, in this study, Fengqiu provided more hydrogen with higher humidity than Hemuqiao despite the occurrence of more frequent precipitation and greater total precipitation at the Hemuqiao station. Differences in SWC storage between Fengqiu and Hemuqiao are most likely predominantly influenced by topography. Thin soil

thickness and a large terrain gradient characterize Hemuqiao [28], while, in Fengqiu, flat farmland, which retains more soil water than the land in Hemuqiao, can provide abundant hydrogen resources.

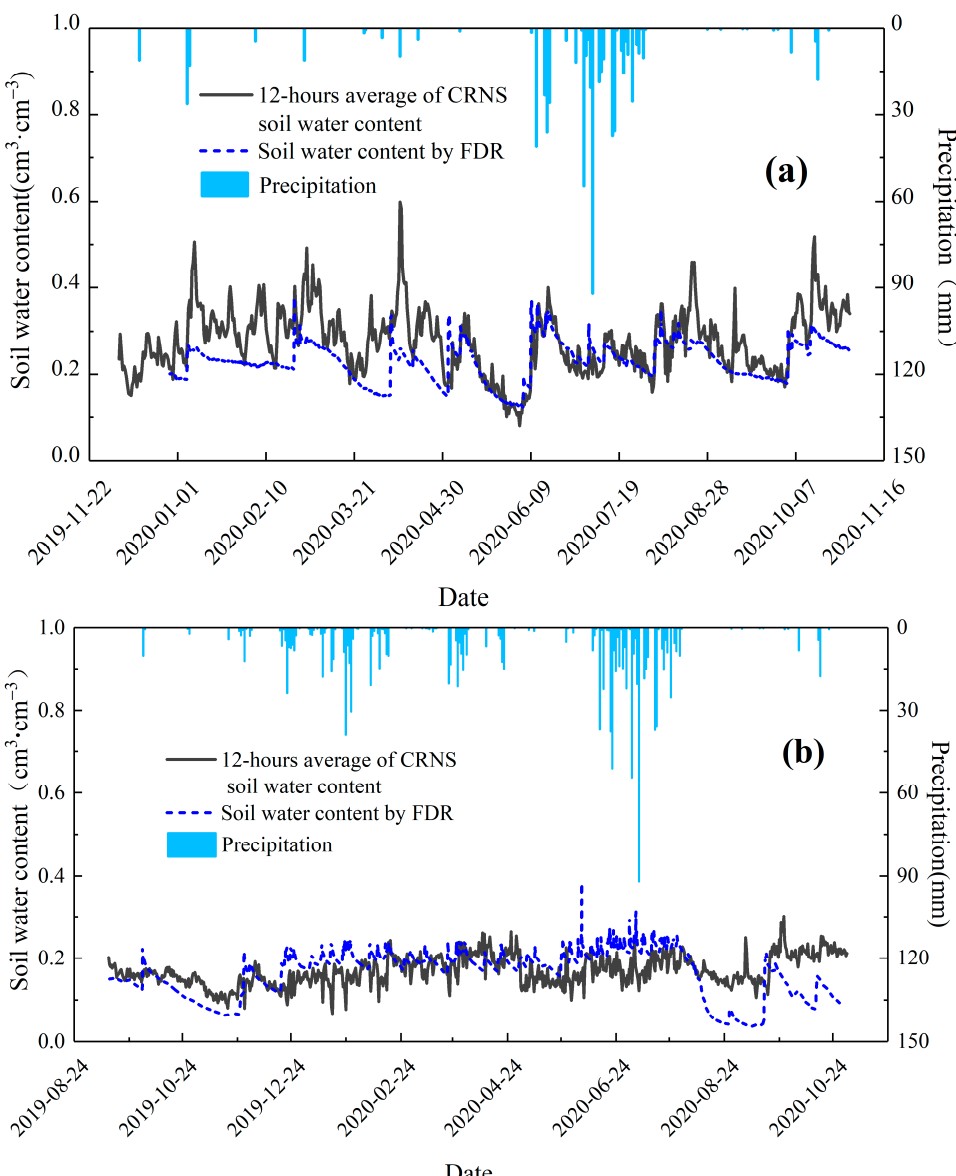

**Figure 3.** Comparison of SWC measured by the CRNS and the FDR at the Fengqiu station (**a**) and the Hemuqiao station (**b**). $N_0$ corresponds to the neutron count rate on 4 June 2020, at the Fengqiu station, while $N_0$ corresponds to the neutron count rate on 18 November 2019, at the Hemuqiao station.

Seasonal differences in SWC time series measured by the CRNS were observed at the two study sites. For most of our study period at the Fengqiu station, SWC measured by CRNS was slightly higher than that in the SWC time series. The diversity was even more obvious in the arid climatic conditions that dominate from January to April, when precipitation is relatively rare. SWC measured by CRNS was also sensitive to other factors, such as vegetation growth and atmospheric pressure [16,27]. The climatic conditions and crop growth may have decreased the neutron count accuracy and precision. The difference in SWC time series suggested that the CRNS captured the changes in both SWC and other hydrogen sources. However, the period from August to November at the Hemuqiao station is arid because of the low precipitation (see Figure 3b), and SWC measured by CRNS decreased as SWC decreased. The time series of SWC measured by CRNS was less volatile

and lower than the SWC measured at the Hemuqiao station, except during the arid periods. The SWC measured by the CRNS relied on the neutron count rates of the CRNP. Neutrons are transported in the soil over distances of a few tens of centimetres [11] and, therefore, were blocked by ridges on the hillslope. Particular mountain terrain caused the radius of the footprint received by the CRNP at the Hemuqiao station to be smaller than that of the URANOS simulation. The narrow neutron fluctuation range measured by CRNS led to the narrow time series fluctuation range in Hemuqiao.

The time series of daily precipitation during the study period is also shown in Figure 3. Distinct seasonality in precipitation was observed at both study sites due to the monsoon climate. The total precipitation during the study period for Fengqiu and Hemuqiao was 647.4 mm and 1155.2 mm, respectively. The rare precipitation and higher SWC of Fengqiu also indicated that farmland has a higher capacity to store water than mountainous regions. Transient changes in SWC generated by precipitation were sensitively observed by CRNS at the Fengqiu station (see Figure 3a). The CRNS accurately captured each precipitation event, although some precipitation events did not have a significant effect on the SWC. The response characteristics to precipitation indicated that CRNS was more sensitive to changes in SWC than other soil-mounted detectors [18,29].

### 4.3. Spatial Characteristics of SWC at Different Study Sites

In this study, the data from the sampling points were calibrated by the weighting function (Sections 3.3 and 3.4) for comparison with CRNS. In addition, direct averaging of sampling points was considered and used for estimating the SWC (see Figure 4b,d). With direct averaging of sampling points, the $R^2$ values were 0.487 and 0.798 for Fengqiu and Hemuqiao, respectively. The RMSE values were 0.054 and 0.049 $cm^3 \cdot cm^{-3}$ for Fengqiu and Hemuqiao, respectively. The weighting calculation improved the accuracy of CRNS estimation in Fengqiu compared to the average calculation, indicating that the weighting simulation was consistent with the spatial characteristics of the CRNS footprint. Previous studies reported that the neutron density detected by CRNP is heterogeneously distributed within the CRNS footprint [30–32]. Despite this result, the weighting calculation did not significantly improve the CRNS estimation accuracy of Hemuqiao.

The linear positive correlation between the SWC measured by CRNS and the SWC at the sampling points indicated that CRNS is capable of accurately estimating the SWC. The estimation accuracy was determined by the coefficient of determination ($R^2$) and the RMSE. In the context of 6 samplings, the $R^2$ for Fengqiu was 0.645, while the RMSE was 0.049 $cm^3 \cdot cm^{-3}$; in the context of 7 samplings, the $R^2$ and RMSE for Hemuqiao were 0.773 and 0.049 $cm^3 \cdot cm^{-3}$, respectively (see Figure 4a,c). The estimation accuracy of CRNS increased as the sampling frequency increases, indicating that CRNS was well correlated with SWC [33,34]. Although the $R^2$ for Fengqiu was lower than that of Hemuqiao, the linear fitting curve for Fengqiu was closer to the 1:1 line than that of Hemuqiao, suggesting that CRNS estimated a more accurate SWC at the Fengqiu station.

It is important to note that topographical differences between the two experimental sites affected the neutron footprint. The distance that neutrons travelled before detection was reduced by the obstruction of the mountain, according to Kohli et al. and Franz et al. [12,24]. The CRNP was installed in the middle of the hillslope at the Hemuqiao station, which was in the saddle of the valley. Therefore, the flanking hillsides blocked part of the neutron footprint, narrowing the actual detection footprint of the CRNS to the region inside the valley. The obstruction of neutron transmission received by CRNS in the horizontal direction could lead to an unacceptable level of precision for SWC estimation in mountainous terrain [35].

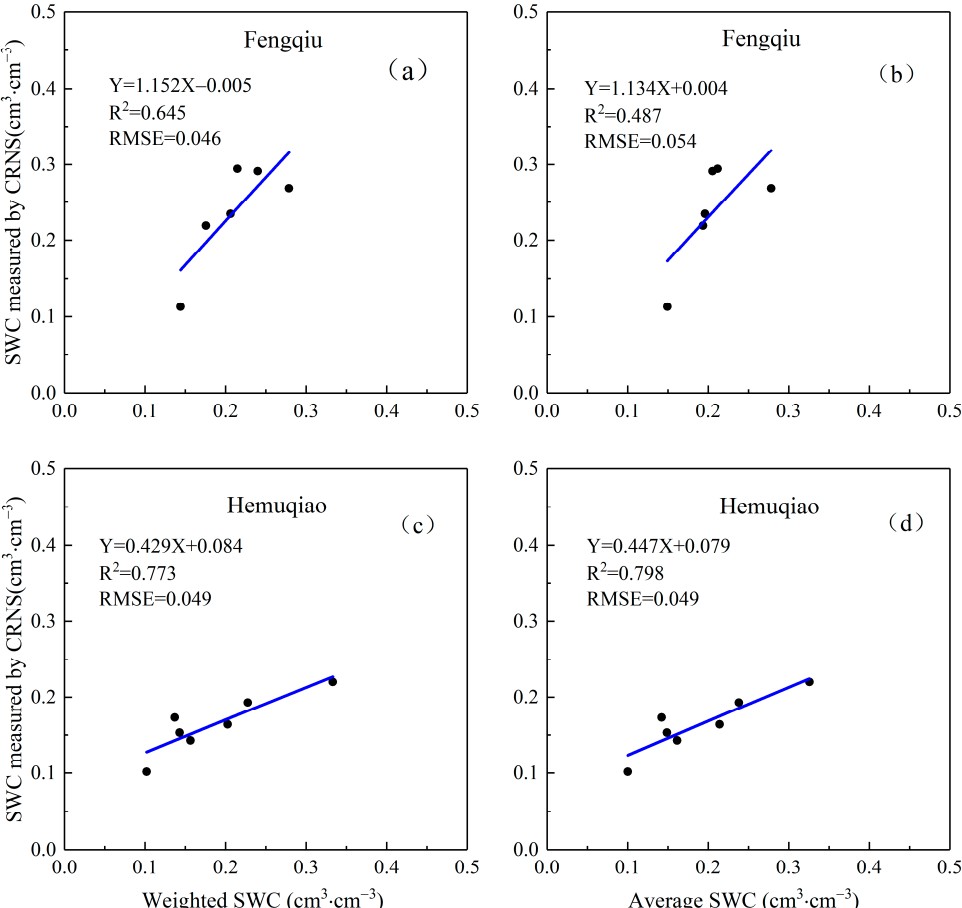

**Figure 4.** The SWC correlation between CRNS and point sampling. (**a**) The SWC correlation between CRNS and weighted point sampling in Fengqiu. (**b**) The SWC correlation between CRNS and averaged point sampling in Fengqiu. (**c**) The SWC correlation between CRNS and weighted point sampling in Hemuqiao. (**d**) The SWC correlation between CRNS and averaged point sampling in Hemuqiao.

The time series variation of CRNS is an accurate indicator of the accuracy of CRNS estimates. Significant differences between the two experimental sites in the correlation daily variations of CRNS and FDR were observed in Figure 5. At Fengqiu, the determination coefficient between CRNS and FDR was 0.390 with an RMSE of 0.072 cm$^3$·cm$^{-3}$. However, no significant correlation between the daily variation in CRNS and FDR was observed at Hemuqiao. The time series of SWC measured by FDR represented a continuous change in SWC at the experimental site. Therefore, the determination coefficient of 0.059 indicated that CRNS could not accurately represent the diurnal variation in SWC at Hemuqiao. CRNS counts depend on the proximity of neutron density to the surface, and thus fluctuate within a certain range. Increasing the sampling interval can improve statistical stability, as previously reported in several studies [21,33]. This study assumed that the daily or 12 h variation in the CRNS could adequately describe the temporal variation in SWC without increasing the error due to long sampling intervals.

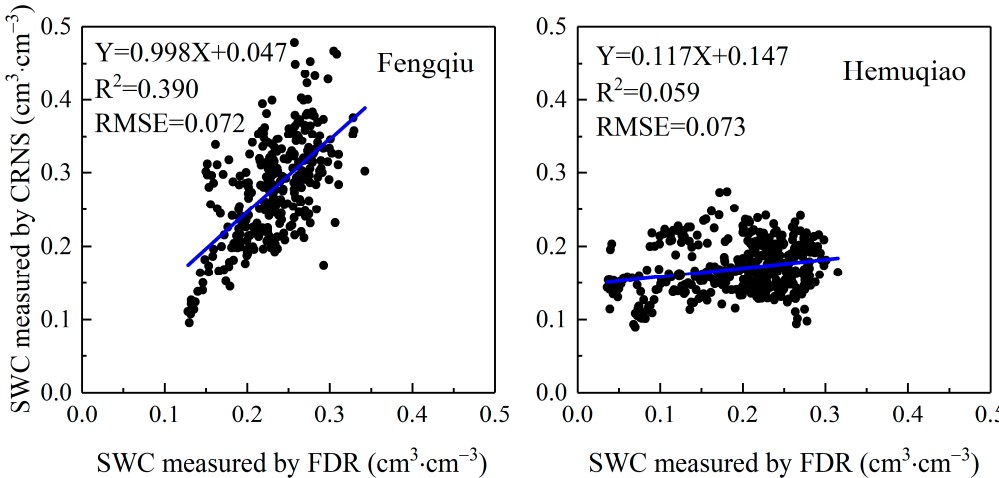

**Figure 5.** Correlation between the daily variations in SWC measured by CRNS and FDR.

The vegetation factor at Hemuqiao was also nonnegligible for the accuracy of CRNS estimates. The pole height of the moso bamboo is over 20 m. Most of its canopy extends above the CRNP, which is the largest difference between forest vegetation and farm vegetation. Fortunately, the hydrogen content of moso bamboo is assumed to be constant, and only the leaves change seasonally. Research has shown that, during the wet season, the hydrogen storage in tree canopies and litter layers can reach 0.067 $cm^3 \cdot cm^{-3}$ of soil volumetric water content [20]. However, it is not possible to determine the dynamic changes in bamboo forest contributions to the hydrogen pool during wet conditions with high interception storage. Therefore, we recommend that calibration for forest vegetation be considered in further studies to obtain more accurate estimates.

## 5. Conclusions

This study applied CRNS to estimate the SWC at two different experimental sites: a typical monsoonal farmland and a humid mountainous region. The URANOS simulations of the footprint and weighting functions show that the detection radius of CRNS ranges between 125 m and 153 m for different humidity conditions in mountainous regions, and the detection depth of CRNS ranges between 19.1 cm and 45.9 cm for different humidity conditions. In farmland, the detection radius of CRNS ranges from 127 m to 139 m, and the detection depth ranges from 16.9 cm to 31.2 cm depending on different humidity conditions. In general, farmland has a smaller CRNS footprint than mountainous regions due to its flat topography and thicker soils, which can retain more hydrogen resources.

The SWC measured by CRNS is generally slightly higher than that measured by the soil-mounted detectors, which is likely due to the influence of hydrogen sources other than the soil. Furthermore, CRNS is more sensitive to changes in SWC than other soil-mounted detectors, suggesting that correction of nonsoil hydrogen sources should be considered when monitoring SWC with CRNS.

The comparison between the CRNS and point sampling estimates show that the estimation accuracy varies between the two different experimental sites, but the difference is not statistically significant. The use of weighting calculation in farmland significantly improves the estimation accuracy of CRNS, indicating the necessity of using spatial weighting simulations for CRNS in order to obtain more accurate results. However, the accuracy of SWC estimation using CRNS in mountainous areas cannot be further improved by using weighting calculations. Previous research has indicated that the use of CRNS is suitable for estimating the SWC of farmland; however, there are many uncertainties when applying CRNS to estimate SWC in mountainous areas due to the terrain obstructing neutron transmission and tall vegetation introducing errors into the CRNS estimation results. Future studies should pay greater attention to the effects of topography and vegetation when CRNS is applied in mountainous environments.

**Author Contributions:** Conceptualization, Y.J., X.L. and J.L.; methodology, Y.J., X.L. and J.L.; software, Y.J. and K.X.; validation, Y.J. and C.G.; investigation, Y.J., Y.Z. and Y.L.; resources, Y.J., H.D. and K.X.; data curation, Y.J.; writing—original draft preparation, Y.J.; writing—review and editing, Y.J. and J.L.; supervision, J.L. and X.L.; funding acquisition, J.L. and X.L. All authors have read and agreed to the published version of the manuscript.

**Funding:** This work was supported by the National Key Research and Development Program of China (Funding number, 2022YFD1500502), the Strategic Priority Research Program of the Chinese Academy of Sciences (grant No. XDA28010401), and the National Natural Science Foundation of China (grant Nos. 42177302 and 41877021).

**Data Availability Statement:** The data presented in this study are available on request from the corresponding author.

**Acknowledgments:** We would like to express our appreciation to the Fengqiu State Key Argo-Ecological Experimental Station and Hemuqiao Experimental Station for providing the experimental site support for this research.

**Conflicts of Interest:** The authors declare no conflict of interest.

## Appendix A. Detection Radius (m) of Simulations at Different SWCs ($cm^3 \cdot cm^{-3}$) and air Humidities ($g \cdot cm^{-2}$)

**Table A1.** Detection radius obtained by simulations under different SWCs ($cm^3 \cdot cm^{-3}$) and air humidities ($g \cdot cm^{-2}$). The pattern of variations in the detection radius values indicates the impact of SWC and air humidity on detection radius.

| SWC | 0.10 | 0.15 | 0.20 | 0.25 | 0.30 | 0.35 | 0.40 | 0.45 |
|---|---|---|---|---|---|---|---|---|
| Radius with the air humidity of 10 | 187 | 178 | 172 | 166 | 161 | 157 | 153 | 150 |
| Air humidity | 5 | 10 | 15 | 20 | 25 | 30 | 33 | |
| Radius with SWC of 0.15 | 184 | 178 | 173 | 168 | 164 | 160 | 158 | |

Note: Multiple regression analysis showed that both air humidity and SWC had a significant impact on the detection radius ($p < 0.001$), and the adjusted $R^2$ was 0.976.

## Appendix B. Detection Depth (cm) of Simulations at Different SWCs ($cm^3 \cdot cm^{-3}$) and air Humidities ($g \cdot cm^{-2}$)

**Table A2.** Detection depth obtained by simulations under different SWCs ($cm^3 \cdot cm^{-3}$) and air humidities ($g \cdot cm^{-2}$). The pattern of variations in the detection depth values indicates the impact of SWC and air humidity on detection depth.

| SWC | 0.10 | 0.15 | 0.20 | 0.25 | 0.30 | 0.35 | 0.40 | 0.45 |
|---|---|---|---|---|---|---|---|---|
| Depth with the air humidity of 10 | 35.3 | 29.2 | 24.8 | 21.6 | 19.1 | 16.9 | 15.2 | 13.7 |
| Air humidity | 5 | 10 | 15 | 20 | 25 | 30 | 33 | |
| Depth with SWC of 0.15 | 29.1 | 29.2 | 29.2 | 29.2 | 29.3 | 29.4 | 29.4 | |

Note: Multiple regression analysis was used to investigate the effects of air humidity and SWC on the detection depth, and the adjusted $R^2$ was 0.954. SWC had a significant impact on the detection depth ($p < 0.001$), but there was no significant impact observed on air humidity to the detection depth ($p = 0.919$).

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
