# Peer review of "Investigating the Potential of Cosmic-Ray Neutron Sensing for Estimating Soil Water Content in Farmland and Mountainous Areas"

_water, doi:10.3390/w15081500_

Round 1
Reviewer 1 Report
The manuscript entitled “Investigating the Potential of Cosmic-Ray Neutron Sensing for Estimating Soil Water Content in Farmland and Mountainous Areas” which was submitted in Water Journal has been reviewed gently.
This experiment has enough novelty and data which can be publish in a well-known journal i.e., Water. Furthermore, the manuscript is based on impressive empirical evidence and makes an original contribution. I want to give some suggestions which can increased the quality of the manuscript and attract readership. These comments are as follow:
On the abstract: Please write all of your protocols at-least once in the results sections. It will give an idea that which parameters you have been studied in this experiment.
On the Introduction: Please re-write your objectives in this manner such as “Therefore, we have conducted the present experiment (i) to investigating the Potential of Cosmic-Ray Neutron Sensing for Estimating Soil Water Content in Farmland and Mountainous Areas, etc.
On the M&M. Figure 1 is difficult to read and should be adjusted.
“Figure 1 and 2 can be combined to create a better overview.”
On the results & discussion: Please re-write your 4.2 sub-heading in details. Also, include some more details about the variation pattern was observed at both study sites
On the conclusion: Please write one more sentence about the future recommendation of this study.
Reviewer 2 Report
Explain line 149 "Due to topographical factors, the TDR soil moisture monitor was used in subsequent soil moisture sampling of the study site." What does this mean? What were the factors and why did the lead to TDR monitoring used subsequently?
Line 99: "self-developed" makes it sound like the probe developed itself (a strange image)! Maybe try something like "We developed a comsmic-ray neutron sensing probe (CRNP) for this study using a proportional counter tube filled with BF3, which can detect neutrons ..."
Figure 1 and Figure 2. Indicate which study site is shown in each figure caption.
Please explain why figure 4 has only 6 or 7 sample points per site. What is meant by "average"? Is it averaged over time, over space (multiple sample points) or what? This seems to be unclear. I would have expected about 32 points, one for each of the soil sampling points at each site.
A well-written paper!
Reviewer 3 Report
Dear Authors,
The subject of the paper is a current, very recent and important topic since it is a contribution to the study of soil water content estimation using the Cosmic-Ray Neutron Sensing method. In my opinion, the subject fits the scope of the Water journal.
The aim of the research is to evaluate the effectiveness of CRNS in estimating SWC in flat farmland areas with a warm-temperate, semi humid monsoon climate and steep headwater hillslopes in a subtropical humid climate.
The authors concluded that CRNS is suitable for use in farmland and mountainous areas and that further attention should be given to the effects of topography and vegetation when it is applied in mountainous environments.
Main Remarks
The document is well written but, in my opinion, some parts could be improved. For example:
- The description of the CNRS theory should be clearer, complete and detailed (lines 46-52).
- Pictures of the Cosmic-ray neutron sensing probe and meteorological equipment should be placed in section 2.1.
- lines 101-104: I don't understand why the sentence is written in the past tense.
- lines 338-348: This paragraph should be placed before the previous paragraph.
- Information from the appendices could be inserted into the main text.
